# Finite-Time Performance Bounds and Adaptive Learning Rate Selection for Two Time-Scale Reinforcement Learning

**Harsh Gupta**
ECE and CSL
University of Illinois at Urbana-Champaign
hgupta10@illinois.edu

**R. Srikant**
ECE and CSL
University of Illinois at Urbana-Champaign
rsrikant@illinois.edu

**Lei Ying**
EECS
University of Michigan, Ann Arbor
leiying@umich.edu

## Abstract

We study two time-scale linear stochastic approximation algorithms, which can be used to model well-known reinforcement learning algorithms such as GTD, GTD2, and TDC. We present finite-time performance bounds for the case where the learning rate is fixed. The key idea in obtaining these bounds is to use a Lyapunov function motivated by singular perturbation theory for linear differential equations. We use the bound to design an adaptive learning rate scheme which significantly improves the convergence rate over the known optimal polynomial decay rule in our experiments, and can be used to potentially improve the performance of any other schedule where the learning rate is changed at pre-determined time instants.

## 1 Introduction

A key component of reinforcement learning algorithms is to learn or approximate value functions under a given policy [Sutton, 1988], [Bertsekas and Tsitsiklis, 1996], [Szepesvári, 2010], [Bertsekas, 2011], [Bhatnagar et al., 2012], [Sutton and Barto, 2018]. Many existing algorithms for learning value functions are variants of the temporal-difference (TD) learning algorithms [Sutton, 1988], [Tsitsiklis and Van Roy, 1997], and can be viewed as stochastic approximation algorithms for minimizing the Bellman error (or objectives related to the Bellman error). Characterizing the convergence of these algorithms, such as TD(0), TD($\lambda$), GTD , nonlinear GTD has been an important objective of reinforcement learning [Szepesvári, 2010], [Bhatnagar et al., 2009], and [Sutton et al., 2016]. The asymptotic convergence of these algorithms with diminishing steps has been established using stochastic approximation theory in many prior works (comprehensive surveys on stochastic approximations can be found in [Benveniste et al., 2012], [Kushner and Yin, 2003], and [Borkar, 2009]).

The conditions required for theoretically establishing asymptotic convergence in an algorithm with diminishing step sizes imply that the learning rate becomes very small very quickly. As a result, the algorithm will require a very large number of samples to converge. Reinforcement learning algorithms used in practice follow a pre-determined learning rate (step-size) schedule which, in most cases, uses decaying step sizes first and then a fixed step size. This gap between theory and practice has prompted a sequence of works on finite-time performance of temporal difference learning algorithms with either time-varying step sizes or constant step sizes [Dalal et al., 2017a,b, Liu et al.,

2018, Lakshminarayanan and Szepesvari, 2018, Bhandari et al., 2018, Srikant and Ying, 2019]. Most of these results are for single time-scale TD algorithms, except [Dalal et al., 2017b] which considers two time-scale algorithms with decaying step sizes. Two time-scale TD algorithms are an important class of reinforcement learning algorithms because they can improve the convergence rate of TD learning or remedy the instability of single time-scale TD in some cases. This paper focuses on two time-scale linear stochastic approximation algorithms with constant step sizes. The model includes TDC, GTD and GTD2 as special cases (see [Sutton et al., 2008], [Sutton et al., 2009] and [Szepesvári, 2010] for more details). We note that, in contemporaneous work, [Xu et al., 2019] have carried out a two-time-scale analysis of linear stochastic approximation with diminishing step sizes.

Besides the theoretical analysis of finite-time performance of two time-scale reinforcement learning algorithms, another important aspect of reinforcement learning algorithms, which is imperative in practice but has been largely overlooked, is the design of learning rate schedule, i.e., how to choose proper step-sizes to improve the learning accuracy and reduce the learning time. This paper addresses this important question by developing principled heuristics based on the finite-time performance bounds.

The main contributions of this paper are summarized below.

- **Finite Time Performance Bounds:** We study two time-scale linear stochastic approximation algorithms, driven by Markovian samples. We establish finite time bounds on the mean-square error with respect to the fixed point of the corresponding ordinary differential equations (ODEs). The performance bound consists of two parts: a steady-state error and a transient error, where the steady-state error is determined by the step sizes but independent of the number of samples (or number of iterations), and the transient error depends on both step sizes and the number of samples. The transient error decays geometrically as the number of samples increases. The key differences between this paper and [Dalal et al., 2017b] include (i) we do not require a sparse projection step in the algorithm; and (ii) we assume constant step-sizes which allows us to develop the adaptive step-size selection heuristic mentioned next.

- **Adaptive Learning Rate Selection:** Based on the finite-time performance bounds, in particular, the steady-state error and the transient error terms in the bounds, we propose an adaptive learning rate selection scheme. The intuition is to use a constant learning rate until the transient error is dominated by the steady-state error; after that, running the algorithm further with the same learning rate is not very useful and therefore, we reduce the learning rate at this time. To apply adaptive learning rate selection in a model-free fashion, we develop data-driven heuristics to determine the time at which the transient error is close to the steady-state error. A useful property of our adaptive rate selection scheme is that it can be used with any learning rate schedule which already exists in many machine learning software platforms: one can start with the initial learning rate suggested by such schedules and get improved performance by using our adaptive scheme. Our experiments on Mountain Car and Inverted Pendulum show that our adaptive learning rate selection significantly improves the convergence rates as compared to optimal polynomial decay learning rate strategies (see [Dalal et al., 2017b] and [Konda et al., 2004] for more details on polynomial decay step-size rules).

## 2    Model, Notation and Assumptions

We consider the following two time-scale linear stochastic approximation algorithm:

$$
\begin{aligned}
U_{k+1} &= U_k + \epsilon^\alpha \left( A_{uu}(X_k)U_k + A_{uv}(X_k)V_k + b_u(X_k) \right) \\
V_{k+1} &= V_k + \epsilon^\beta \left( A_{vu}(X_k)U_k + A_{vv}(X_k)V_k + b_v(X_k) \right),
\end{aligned}
\tag{1}
$$

where $\{X_k\}$ are the samples from a Markov process. We assume $\beta < \alpha$ so that, over $\epsilon^{-\beta}$ iterations, the change in $V$ is $O(1)$ while the change in $U$ is $O\left(\epsilon^{\alpha-\beta}\right)$. Therefore, $V$ is updated at a faster time scale than $U$.

In the context of reinforcement learning, when combined with linear function approximation of the value function, GTD, GTD2, and and TDC can be viewed as two time-scale linear stochastic approximation algorithms, and can be described in the same form as (1). For example, GTD2 with

linear function approximation is as follows:

$$U_{k+1} = U_k + \epsilon^\alpha \left( \phi(X_k) - \zeta\phi(X_{k+1}) \right) \phi^\top(X_k) V_k$$

$$V_{k+1} = V_k + \epsilon^\beta \left( \delta_k - \phi^\top(X_k) V_k \right) \phi(X_k),$$

where $\zeta$ is the discount factor, $\phi(x)$ is the feature vector of state $x$, $U_k$ is the weight vector such that $\phi^\top(x)U_k$ is the approximation of value function of state $x$ at iteration $k$, $\delta_k = c(X_k) + \zeta\phi^\top(X_{k+1})U_k - \phi^\top(X_k)U_k$ is the TD error, and $V_k$ is the weight vector that estimates $\mathbb{E}[\phi(X_k)\phi(X_k)^T]^{-1}\mathbb{E}[\delta_k\phi(X_k)]$.

We now summarize the notation we use throughout the paper and the assumptions we make.

- **Assumption 1:** $\{X_k\}$ is a Markov chain with state space $\mathcal{S}$. We assume that the following two limits exist:

$$\begin{pmatrix} \bar{A}_{uu} & \bar{A}_{uv} \\ \bar{A}_{vu} & \bar{A}_{vv} \end{pmatrix} = \lim_{k\to\infty} \begin{pmatrix} \mathbb{E}\left[A_{uu}(X_k)\right] & \mathbb{E}\left[A_{uv}(X_k)\right] \\ \mathbb{E}\left[A_{vu}(X_k)\right] & \mathbb{E}\left[A_{vv}(X_k)\right] \end{pmatrix}$$

$$\begin{pmatrix} \bar{b}_u & \bar{b}_v \end{pmatrix} = \lim_{k\to\infty} \begin{pmatrix} \mathbb{E}[b_u(X_k)] & \mathbb{E}[b_v(X_k)] \end{pmatrix} = 0.$$

Note that without the loss of generality, we assume $\bar{b} = 0$ which allows for the fixed point of the associated ODEs to be 0. This can be guaranteed by appropriate centering. We define

$$B(X_k) = A_{uu}(X_k) - A_{uv}(X_k)\bar{A}_{vv}^{-1}\bar{A}_{vu} \qquad \tilde{B}(X_k) = A_{vu}(X_k) - A_{vv}(X_k)\bar{A}_{vv}^{-1}\bar{A}_{vu}$$

$$\bar{B} = \bar{A}_{uu} - \bar{A}_{uv}\bar{A}_{vv}^{-1}A_{vu} \qquad \bar{\tilde{B}} = \bar{A}_{vu} - \bar{A}_{vv}\bar{A}_{vv}^{-1}\bar{A}_{vu}.$$

- **Assumption 2:** We assume that $\max\{\|b_u(x)\|, \|b_v(x)\|\} \leq b_{\max} < \infty$ for any $x \in \mathcal{S}$. We also assume that $\max\{\|B(x)\|, \|\tilde{B}(x)\|, \|A_{uu}(x)\|, \|A_{vu}(x)\|, \|A_{uv}(x)\|, \|A_{vv}(x)\|\} \leq 1$ for any $x \in \mathcal{S}$. Note that these assumptions imply that the steady-state limits of the random matrices/vectors will also satisfy the same inequalities.

- **Assumption 3:** We assume $\bar{A}_{vv}$ and $\bar{B}$ are Hurwitz and $\bar{A}_{vv}$ is invertible. Let $P_u$ and $P_v$ be the solutions to the following Lyapunov equations:

$$-I = \bar{B}^\top P_u + P_u\bar{B}$$

$$-I = \bar{A}_{vv}^\top P_v + P_v\bar{A}_{vv}.$$

Since both $\bar{A}_{vv}$ and $\bar{B}$ are Hurwitz, $P_u$ and $P_v$ are real positive definite matrices.

- **Assumption 4:** Define $\tau_\Delta \geq 1$ to be the mixing time of the Markov chain $\{X_k\}$. We assume

$$\|\mathbb{E}[b_k|X_0 = i]\| \leq \Delta, \forall i, \forall k \geq \tau_\Delta$$

$$\|\bar{B} - \mathbb{E}[B(X_k)|X_0 = i]\| \leq \Delta, \forall i, \forall k \geq \tau_\Delta$$

$$\|\bar{\tilde{B}} - \mathbb{E}[\tilde{B}(X_k)|X_0 = i]\| \leq \Delta, \forall i, \forall k \geq \tau_\Delta$$

$$\|\bar{A}_{uv} - \mathbb{E}[A_{uv}(X_k)|X_0 = i]\| \leq \Delta, \forall i, \forall k \geq \tau_\Delta$$

$$\|\bar{A}_{vv} - \mathbb{E}[A_{vv}(X_k)|X_0 = i]\| \leq \Delta, \forall i, \forall k \geq \tau_\Delta.$$

- **Assumption 5:** As in [Srikant and Ying, 2019], we assume that there exists $K \geq 1$ such that $\tau_\Delta \leq K \log(\frac{1}{\Delta})$. For convenience, we choose

$$\Delta = 2\epsilon^\alpha \left( 1 + \|\bar{A}_{vv}^{-1}\bar{A}_{vu}\| + \epsilon^{\beta-\alpha} \right)$$

and drop the subscript from $\tau_\Delta$, i.e., $\tau_\Delta = \tau$. Also, for convenience, we assume that $\epsilon$ is small enough such that $\tilde{\epsilon}\tau \leq \frac{1}{4}$, where $\tilde{\epsilon} = \Delta = 2\epsilon^\alpha \left( 1 + \|\bar{A}_{vv}^{-1}\bar{A}_{vu}\| + \epsilon^{\beta-\alpha} \right)$.

We further define the following notation:

- Define matrix

$$P = \begin{pmatrix} \frac{\xi_v}{\xi_u+\xi_v}P_u & 0 \\ 0 & \frac{\xi_u}{\xi_u+\xi_v}P_v \end{pmatrix}, \tag{2}$$

where $\xi_u = 2\|P_u\bar{A}_{uv}\|$ and $\xi_v = 2\left\|P_v\bar{A}_{vv}^{-1}\bar{A}_{vu}\bar{B}\right\|$.

- Let $\gamma_{\max}$ and $\gamma_{\min}$ denote the largest and smallest eigenvalues of $P_u$ and $P_v$, respectively. So $\gamma_{\max}$ and $\gamma_{\min}$ are also upper and lower bounds on the eigenvalues of $P$.

## 3 Finite-Time Performance Bounds

To establish the finite-time performance guarantees of the two time-scale linear stochastic approximation algorithm (1), we define

$$Z_k = V_k + \bar{A}_{vv}^{-1}\bar{A}_{vu}U_k \quad \text{and} \quad \Theta_k = \begin{pmatrix} U_k \\ Z_k \end{pmatrix}.$$

Then we consider the following Lyapunov function:

$$W(\Theta_k) = \Theta_k^\top P \Theta_k, \tag{3}$$

where $P$ is a symmetric positive definite matrix defined in (2) ($P$ is positive definite because both $P_u$ and $P_v$ are positive definite matrices). The reason to introduce $Z_k$ will become clear when we introduce the key idea of our analysis based on singular perturbation theory.

The following lemma bounds the expected change in the Lyapunov function in one time step.

**Lemma 1.** *For any $k \geq \tau$ and $\epsilon$, $\alpha$, and $\beta$ such that $\eta_1 \tilde{\epsilon} \tau + 2\frac{\tilde{\epsilon}^2}{\epsilon^\alpha}\gamma_{\max} \leq \frac{\kappa_1}{2}$, the following inequality holds:*

$$\mathbb{E}[W(\Theta_{k+1}) - W(\Theta_k)] \leq -\frac{\epsilon^\alpha}{\gamma_{\max}}\left(\frac{\kappa_1}{2} - \kappa_2 \epsilon^{\alpha-\beta}\right)\mathbb{E}[W(\Theta_k)] + \epsilon^{2\beta}\tau\eta_2,$$

*where $\tilde{\epsilon} = 2\epsilon^\alpha\left(1 + \|\bar{A}_{vv}^{-1}\bar{A}_{vu}\| + \epsilon^{\beta-\alpha}\right)$, and $\eta_1$, $\eta_2$ $\kappa_1$, and $\kappa_2$ are constants independent of $\epsilon$.*

The proof of Lemma 1 is somewhat involved, and is provided in the supplementary material. The definitions of $\eta_1$, $\eta_2$, $\kappa_1$ and $\kappa_2$ can be found in the supplementary material as well. Here, we provide some intuition behind the result by studying a related ordinary differential equation (ODE). In particular, consider the expected change in the stochastic system divided by the slow time-scale step size $\epsilon^\alpha$:

$$\frac{\mathbb{E}[U_{k+1} - U_k|U_{k-\tau} = u, V_{k-\tau} = v, X_{k-\tau} = x]}{\epsilon^\alpha}$$
$$=\mathbb{E}\left[\left(A_{uu}(X_k)U_k + A_{uv}(X_k)V_k + b_u\right)|U_{k-\tau} = u, V_{k-\tau} = v, X_{k-\tau} = x\right] \tag{4}$$
$$\epsilon^{\alpha-\beta}\frac{\mathbb{E}[V_{k+1} - V_k|U_{k-\tau} = u, V_{k-\tau} = v, X_{k-\tau} = x]}{\epsilon^\alpha}$$
$$=\mathbb{E}\left[\left(A_{vu}(X_k)U_k + A_{vv}(X_k)V_k + b_v(X_k)\right)|U_{k-\tau} = u, V_{k-\tau} = v, X_{k-\tau} = x\right],$$

where the expectation is conditioned sufficiently in the past in terms of the underlying Markov chain (i.e. conditioned on the state at time $k - \tau$ instead of $k$) so the expectation is approximately in steady-state.

Approximating the left-hand side by derivatives and the right-hand side using steady-state expectations, we get the following ODEs:

$$\dot{u} = \bar{A}_{uu}u + \bar{A}_{uv}v \tag{5}$$
$$\epsilon^{\alpha-\beta}\dot{v} = \bar{A}_{vu}u + \bar{A}_{vv}v. \tag{6}$$

Note that, in the limit as $\epsilon \to 0$, the second of the above two ODEs becomes an algebraic equation, instead of a differential equation. In the control theory literature, such systems are called singularly-perturbed differential equations, see for example [Kokotovic et al., 1999]. In [Khalil, 2002, Chapter 11], the following Lyapunov equation has been suggested to study the stability of such singularly perturbed ODEs:

$$W(u,v) = du^\top P_u u + (1-d)\left(v + \bar{A}_{vv}^{-1}\bar{A}_{vu}u\right)^\top P_v\left(v + \bar{A}_{vv}^{-1}\bar{A}_{vu}u\right), \tag{7}$$

for $d \in [0, 1]$. The function $W$ mentioned earlier in (3) is the same as above for a carefully chosen $d$. The rationale behind the use of the Lyapunov function (7) is presented in the appendix.

The intuition behind the result in Lemma 1 can be understood by studying the dynamics of the above Lyapunov function in the ODE setting. To simplify the notation, we define $z = v + \bar{A}_{vv}^{-1}\bar{A}_{vu}u$, so the Lyapunov function can also be written as

$$W(u,z) = du^\top P_u u + (1-d)z^\top P_v z, \tag{8}$$

and adapting the manipulations for nonlinear ODEs in [Khalil, 2002, Chapter 11] to our linear model, we get

$$\dot{W} = 2du^T P_u \dot{u} + 2(1-d)z^\top P_v \dot{z} \tag{9}$$

$$\leq - \left(\|u\| \quad \|z\|\right) \tilde{\Psi} \begin{pmatrix} \|u\| \\ \|z\| \end{pmatrix}, \tag{10}$$

where

$$\tilde{\Psi} = \begin{pmatrix} d & -d\gamma_{\max} - (1-d)\gamma_{\max}\sigma_{\min} \\ -d\gamma_{\max} - (1-d)\gamma_{\max}\sigma_{\min} & \left(\frac{1-d}{2\epsilon^{\alpha-\beta}} - (1-d)\gamma_{\max}\sigma_{\min}\right) \end{pmatrix}. \tag{11}$$

Note that $\tilde{\Psi}$ is positive definite when

$$d\left(\frac{1-d}{2\epsilon^{\alpha-\beta}} - (1-d)\gamma_{\max}\sigma_{\min}\right) \geq \left(d\gamma_{\max} + (1-d)\gamma_{\max}\sigma_{\min}\right)^2, \tag{12}$$

i.e., when

$$\epsilon^{\alpha-\beta} \leq \frac{d(1-d)}{2d(1-d)\gamma_{\max}\sigma_{\min} + \left(d\gamma_{\max} + (1-d)\gamma_{\max}\sigma_{\min}\right)^2}. \tag{13}$$

Let $\tilde{\lambda}_{\min}$ denote the smallest eigenvalue of $\tilde{\Psi}$. We have

$$\dot{W} \leq -\tilde{\lambda}_{\min}\left(\|u\|^2 + \|z\|^2\right) \leq -\frac{\tilde{\lambda}_{\min}}{\gamma_{\max}}W. \tag{14}$$

In particular, recall that we obtained the ODEs by dividing by the step-size $\epsilon^\alpha$. Therefore, for the discrete equations, we would expect

$$\mathbb{E}[W(\Theta_{k+1}) - W(\Theta_k)] \approx \leq -\epsilon^\alpha \frac{\tilde{\lambda}_{\min}}{\gamma_{\max}} \mathbb{E}[W(\Theta_k)], \tag{15}$$

which resembles the transient term of the upper bound in Lemma 1. The exact expression in the discrete, stochastic case is of course different and additionally includes a steady-state term, which is not captured by the ODE analysis above.

Now, we are ready to the state the main theorem.

**Theorem 1.** *For any $k \geq \tau$, $\epsilon$, $\alpha$ and $\beta$ such that $\eta_1 \tilde{\epsilon}\tau + 2\frac{\tilde{\epsilon}^2}{\epsilon^\alpha}\gamma_{\max} \leq \frac{\kappa_1}{2}$, we have*

$$\mathbb{E}[\|\Theta_k\|^2] \leq \frac{\gamma_{\max}}{\gamma_{\min}}\left(1 - \frac{\epsilon^\alpha}{\gamma_{\max}}\left(\frac{\kappa_1}{2} - \kappa_2\epsilon^{\alpha-\beta}\right)\right)^{k-\tau}(1.5\|\Theta_0\| + 0.5b_{\max})^2$$

$$+ \epsilon^{2\beta-\alpha}\frac{\gamma_{\max}}{\gamma_{\min}}\frac{\eta_2\tau}{\left(\frac{\kappa_1}{2} - \kappa_2\epsilon^{\alpha-\beta}\right)}.$$

*Proof.* Applying Lemma 1 recursively, we obtain

$$\mathbb{E}[W(\Theta_k)] \leq u^{k-\tau}\mathbb{E}[W(\Theta_\tau)] + v\frac{1 - u^{k-\tau}}{1-u} \leq u^{k-\tau}\mathbb{E}[W(\Theta_k)] + v\frac{1}{1-u} \tag{16}$$

where $u = 1 - \frac{\epsilon^\alpha}{\gamma_{\max}}\left(\frac{\kappa_1}{2} - \kappa_2\epsilon^{\alpha-\beta}\right)$ and $v = \eta_2\tau\epsilon^{2\beta}$. Also, we have that

$$\mathbb{E}[\|\Theta_k\|^2] \leq \frac{1}{\gamma_{\min}}\mathbb{E}[W(\Theta_k)] \leq \frac{1}{\gamma_{\min}}u^{k-\tau}\mathbb{E}[W(\Theta_\tau)] + v\frac{1}{\gamma_{\min}(1-u)}. \tag{17}$$

Furthermore,

$$\mathbb{E}[W(\Theta_\tau)] \leq \gamma_{\max}\mathbb{E}[\|\Theta_\tau\|^2] \leq \gamma_{\max}\mathbb{E}[(\|\Theta_\tau - \Theta_0\| + \|\Theta_0\|)^2]$$

$$\leq \gamma_{\max}\left((1 + 2\tilde{\epsilon}\tau)\|\Theta_0\| + 2\tilde{\epsilon}\tau b_{\max}\right)^2. \tag{18}$$

The theorem then holds using the fact that $\tilde{\epsilon}\tau \leq \frac{1}{4}$. $\qquad \square$

Theorem 1 essentially states that the expected error for a two-time scale linear stochastic approximation algorithm comprises two terms: a *transient error* term which decays geometrically with time and a *steady-state error* term which is directly proportional to $\epsilon^{2\beta-\alpha}$ and the mixing time. This characterization of the finite-time error is useful in understanding the impact of different algorithmic and problem parameters on the rate of convergence, allowing the design of efficient techniques such as the adaptive learning rate rule which we will present in the next section.

# 4 Adaptive Selection of Learning Rates

Equipped with the theoretical results from the previous section, one interesting question that arises is the following: *given a time-scale ratio $\lambda = \frac{\alpha}{\beta}$, can we use the finite-time performance bound to design a rule for adapting the learning rate to optimize performance?*

In order to simplify the discussion, let $\epsilon^\beta = \mu$ and $\epsilon^\alpha = \mu^\lambda$. Therefore, Theorem 1 can be simplified and written as

$$
\mathbb{E}[\|\Theta_k\|^2] \le K_1 \left( 1 - \mu^\lambda \left( \frac{\kappa_1}{2\gamma_{\max}} - \frac{\kappa_2}{\gamma_{\max}} \mu^{\lambda-1} \right) \right)^k + \mu^{2-\lambda} \frac{K_2}{\left( \frac{\kappa_1}{2} - \kappa_2 \mu^{\lambda-1} \right)} \tag{19}
$$

where $K_1$ and $K_2$ are problem-dependent positive constants. Since we want the system to be stable, we will assume that $\mu$ is small enough such that $\frac{\kappa_1}{2\gamma_{\max}} - \frac{\kappa_2}{\gamma_{\max}} \mu^{\lambda-1} = c > 0$. Plugging this condition in (19), we get

$$
\mathbb{E}[\|\Theta_k\|^2] \le K_1 \left( 1 - c\mu^\lambda \right)^k + \frac{K_2 \mu^{2-\lambda}}{\gamma_{\max} c} \tag{20}
$$

In order to optimize performance for a given number of samples, we would like to choose the learning rate $\mu$ as a function of the time step. In principle, one can assume time-varying learning rates, derive more general mean-squared error expressions (similar to Theorem 1), and then try to optimize over the learning rates to minimize the error for a given number of samples. However, this optimization problem is computationally intractable. We note that even if we assume that we are only going to change the learning rate a finite number of times, the resulting optimization problem of finding the times at which such changes are performed and finding the learning rate at these change points is an equally intractable optimization problem. Therefore, we have to devise simpler adaptive learning rate rules.

To motivate our learning rate rule, we first consider a time $T$ such that errors due to the transient and steady-state parts in (20) are equal, i.e.,

$$
K_1(1 - c\mu^\lambda)^T = \frac{K_2 \mu^{2-\lambda}}{\gamma_{\max} c} \tag{21}
$$

From this time onwards, running the two time-scale stochastic approximation algorithm any further with $\mu$ as the learning rate is not going to significantly improve the mean-squared error. In particular, the mean-squared error beyond this time is upper bounded by twice the steady-state error $\frac{K_2 \mu^{2-\lambda}}{\gamma_{\max} c}$. Thus, at time $T$, it makes sense to reset $\mu$ as $\mu \leftarrow \mu/\xi$, where $\xi > 1$ is a hyperparameter. Roughly speaking, $T$ is the time at which one is close to steady-state for a given learning rate, and therefore, it is the time to reduce the learning rate to get to a new "steady-state" with a smaller error.

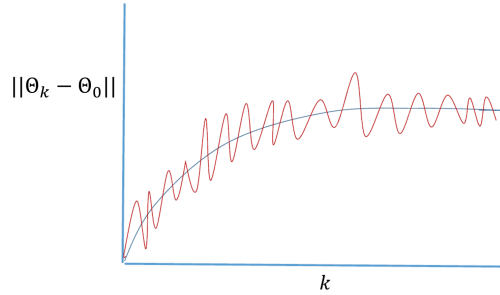

Figure 1: The evolution of $\|\Theta_k - \Theta_0\|$.

The key difficulty in implementing the above idea is that it is difficult to determine $T$. For ease of exposition, we considered a system centered around 0 in our analysis (i.e., $\Theta^* = 0$). More generally, the results presented in Theorem 1 and (19) - (20) will have $\Theta_k$ replaced by $\Theta_k - \Theta^*$. In any practical application, $\Theta^*$ will be unknown. Thus, we cannot determine $\|\Theta_k - \Theta^*\|$ as a function of $k$ and hence, it is difficult to use this approach.

Our idea to overcome this difficulty is to estimate whether the algorithm is close to its steady-state by observing $\|\Theta_k - \Theta_0\|$ where $\Theta_0$ is our initial guess for the unknown parameter vector and is thus known to us. Note that $\|\Theta_k - \Theta_0\|$ is zero at $k = 0$ and will increase (with some fluctuations due to randomness) to $\|\Theta^* - \Theta_0\|$ in steady-state (see Figure 1 for an illustration). Roughly speaking, we approximate the curve in this figure by a sequence of straight lines, i.e., perform a piecewise linear approximation, and conclude that the system has reached steady-state when the lines become approximately horizontal. We provide the details next.

To derive a test to estimate whether $\|\Theta_k - \Theta_0\|$ has reached steady-state, we first note the following inequality for $k \ge T$ (i.e., after the steady-state time defined in (21)):

$$\mathbb{E}[\|\Theta_0 - \Theta^*\|] - \mathbb{E}[\|\Theta_k - \Theta^*\|] \leq \mathbb{E}[\|\Theta_k - \Theta_0\|] \leq \mathbb{E}[\|\Theta_k - \Theta^*\|] + \mathbb{E}[\|\Theta_0 - \Theta^*\|]$$

$$\Rightarrow d - \sqrt{\frac{2K_2\mu^{2-\lambda}}{\gamma_{\max}c}} \leq \mathbb{E}[\|\Theta_k - \Theta_0\|] \leq d + \sqrt{\frac{2K_2\mu^{2-\lambda}}{\gamma_{\max}c}} \tag{22}$$

where the first pair of inequalities follow from the triangle inequality and the second pair of inequalities follow from (20) - (21), Jensen's inequality and letting $d = \mathbb{E}[\|\Theta_0 - \Theta^*\|]$. Now, for $k \geq T$, consider the following $N$ points: $\{X_i = i, Y_i = \|\Theta_{k+i} - \Theta_0\|\}_{i=1}^{N}$. Since these points are all obtained after "steady-state" is reached, if we draw the best-fit line through these points, its slope should be small. More precisely, let $\psi_N$ denote the slope of the best-fit line passing through these $N$ points. Using (22) along with formulas for the slope in linear regression, and after some algebraic manipulations (see Appendix **??** for detailed calculations), one can show that:

$$|\mathbb{E}[\psi_N]| = O\left(\frac{\mu^{1-\frac{\lambda}{2}}}{N}\right), \quad \text{Var}(\psi_N) = O\left(\frac{1}{N^2}\right) \tag{23}$$

Therefore, if $N \geq \frac{\chi}{\mu^{\frac{\lambda}{2}}}$, then the slope of the best-fit line connecting $\{X_i, Y_i\}$ will be $O\left(\frac{\mu^{1-\frac{\lambda}{2}}}{N}\right)$ with high probability (for a sufficiently large constant $\chi > 0$). On the other hand, when the algorithm is in the transient state, the difference between $\|\Theta_{k+m} - \Theta_0\|$ and $\|\Theta_k - \Theta_0\|$ will be $O(m\mu)$ since $\Theta_k$ changes by $O(\mu)$ from one time slot to the next (see Lemma 3 in Appendix **??** for more details). Using this fact, the slope of the best-fit line through $N$ consecutive points in the transient state can be shown to be $O(\mu)$, similar to (23). Since we choose $N \geq \frac{\chi}{\mu^{\frac{\lambda}{2}}}$, the slope of the best-fit line in steady state, i.e., $O\left(\frac{\mu^{1-\frac{\lambda}{2}}}{N}\right)$ will be lower than the slope of the best-fit line in the transient phase, i.e., $O(\mu)$ (for a sufficiently large $\chi$). We use this fact as a diagnostic test to determine whether or not the algorithm has entered steady-state. If the diagnostic test returns true, we update the learning rate (see Algorithm 1).

---

**Algorithm 1** Adaptive Learning Rate Rule

---

**Hyperparameters:** $\rho, \sigma, \xi, N$
Initialize $\mu = \rho$, $\psi_N = 2\sigma\mu^{1-\frac{\lambda}{2}}$, $\Theta_0$, $\Theta_{\text{ini}} = \Theta_0$.
**for** $i = 1, 2, ...$ **do**
    Do two time-scale algorithm update.
    Compute $\psi_N = \text{Slope}\left(\{k, \|\Theta_{i-k} - \Theta_{\text{ini}}\|\}_{k=0}^{N-1}\right)$.
    **if** $\psi_N < \frac{\sigma\mu^{1-\frac{\lambda}{2}}}{N}$ **then**
        $\mu = \frac{\mu}{\xi}$.
        $\Theta_{\text{ini}} = \Theta_i$.
    **end if**
**end for**

---

We note that our adaptive learning rate rule will also work for single time-scale reinforcement learning algorithms such as TD($\lambda$) since our expressions for the mean-square error, when specialized to the case of a single time-scale, will recover the result in [Srikant and Ying, 2019] (see [Gupta et al., 2019] for more details). Therefore, an interesting question that arises from (19) is whether one can optimize the rate of convergence with respect to the time-scale ratio $\lambda$? Since the RHS in (19) depends on a variety of problem-dependent parameters, it is difficult to optimize it over $\lambda$. An interesting direction of further research is to investigate if practical adaptive strategies for $\lambda$ can be developed in order to improve the rate of convergence further.

## 5 Experiments

We implemented our adaptive learning rate schedule on two popular classic control problems in reinforcement learning - Mountain Car and Inverted Pendulum, and compared its performance with the optimal polynomial decay learning rate rule suggested in [Dalal et al., 2017b] (described in the next subsection). See Appendix **??** for more details on the Mountain Car and Inverted Pendulum problems. We evaluated the following policies using the two time-scale TDC algorithm (see [Sutton et al., 2009] for more details regarding TDC):

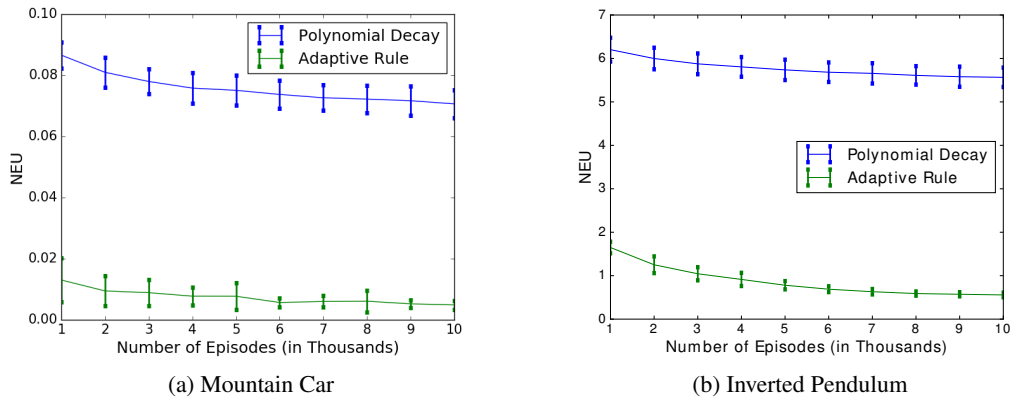

|  | (a) Mountain Car | (b) Inverted Pendulum |

Figure 2: Performance of different learning rate rules in classic control problems.

- Mountain Car - At each time step, choose a random action $\in \{0, 2\}$, i.e., accelerate randomly to the left or right.

- Inverted Pendulum - At each time step, choose a random action in the entire action space, i.e., apply a random torque $\in [-2.0, 2.0]$ at the pivot point.

Since the true value of $\Theta^*$ is not known in both the problems we consider, to quantify the performance of the TDC algorithm, we used the error metric known as the *norm of the expected TD update* (NEU, see [Sutton et al., 2009] for more details). For both problems, we used a $O(3)$ Fourier basis (see [Konidaris et al., 2011] for more details) to approximate the value function and used $0.95$ as the discount factor.

## 5.1 Learning Rate Rules and Tuning

1. The optimal polynomial decay rule suggested in [Dalal et al., 2017b] is the following: at time step $k$, choose $\epsilon_k^\alpha = \frac{1}{(k+1)^\alpha}$ and $\epsilon_k^\beta = \frac{1}{(k+1)^\beta}$, where $\alpha \to 1$ and $\beta \to \frac{2}{3}$. For our experiments, we chose $\alpha = 0.99$ and $\beta = 0.66$. This implies $\lambda = \frac{\alpha}{\beta} = 1.5$. Since the problems we considered require smaller initial step-sizes for convergence, we let $\epsilon_k^\alpha = \frac{\rho_0}{(k+1)^\alpha}$ and $\epsilon_k^\beta = \frac{\rho_0}{(k+1)^\beta}$ and did a grid search to determine the best $\rho_0$, i.e., the best initial learning rate. The following values for $\rho_0$ were found to be the best: *Mountain Car* - $\rho_0 = 0.2$, *Inverted Pendulum* - $\rho_0 = 0.2$.

2. For our proposed adaptive learning rate rule, we fixed $\xi = 1.2$, $N = 200$ in both problems since we did not want the decay in the learning rate to be too aggressive and the resource consumption for slope computation to be high. We also set $\lambda = 1.5$ as in the polynomial decay case to have a fair comparison. We then fixed $\rho$ and conducted a grid search to find the best $\sigma$. Subsequently, we conducted a grid search over $\rho$. Interestingly, the adaptive learning rate rule was reasonably robust to the value of $\rho$. We used $\rho = 0.05$ in Inverted Pendulum and $\rho = 0.1$ in Mountain Car. Effectively, the only hyperparameter that affected the rule's performance significantly was $\sigma$. The following values for $\sigma$ were found to be the best: *Mountain Car* - $\sigma = 0.001$, *Inverted Pendulum* - $\sigma = 0.01$.

## 5.2 Results

For each experiment, one run involved the following: $10,000$ episodes with the number of iterations in each episode being $50$ and $200$ for Inverted Pendulum and Mountain Car respectively. After every $1,000$ episodes, training/learning was paused and the NEU was computed by averaging over $1,000$ test episodes. We initialized $\Theta_0 = 0$. For Mountain Car, $50$ such runs were conducted and the results were computed by averaging over these runs. For Inverted Pendulum, $100$ runs were conducted and the results were computed by averaging over these runs. Note that the learning rate for each adaptive strategy was adapted at the episodic level due to the episodic nature of the problems. The results are reported in Figures 2a and 2b. As is clear from the figures, our proposed adaptive learning rate rule significantly outperforms the optimal polynomial decay rule.

# 6    Conclusion

We have presented finite-time bounds quantifying the performance of two time-scale linear stochastic approximation algorithms. The bounds give insight into how the different time-scale and learning rate parameters affect the rate of convergence. We utilized these insights and designed an adaptive learning rate selection rule. We implemented our rule on popular classical control problems in reinforcement learning and showed that the proposed rule significantly outperforms the optimal polynomial decay strategy suggested in literature.

## Acknowledgements

Research supported by ONR Grant N00014-19-1-2566, NSF Grants CPS ECCS 1739189, NeTS 1718203, CMMI 1562276, ECCS 16-09370, and NSF/USDA Grant AG 2018-67007-28379. Lei Ying's work supported by NSF grants CNS 1618768, ECCS 1609202, IIS 1715385, ECCS 1739344, CNS 1824393 and CNS 1813392.

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
