[Supplementary Material]

# A Proof of Lemma 1

The proof proceeds along similar lines as the corresponding proof in [Srikant and Ying, 2019]. However, the results there cannot be directly applied to get the bounds in this paper due to the fact that we would like to separate out the effects of the $\epsilon$, $\alpha$ and $\beta$ from the other problem parameters, and additionally, the Lyapunov function used here is different.

Recall that
$$Z_k = V_k + \bar{A}_{vv}^{-1} \bar{A}_{vu} U_k,$$
so the stochastic recursions in terms of $(U, Z)$ are
$$U_{k+1} = U_k + \epsilon^\alpha \left( B(X_k) U_k + A_{uv}(X_k) Z_k + b_u(X_k) \right)$$
$$Z_{k+1} = Z_k + \bar{A}_{22}^{-1} \bar{A}_{21} (U_{k+1} - U_k) + \epsilon^\beta \left( \tilde{B}(X_k) U_k + A_{vv}(X_k) Z_k + b_v(X_k) \right)$$
$$= Z_k + \epsilon^\alpha \bar{A}_{22}^{-1} \bar{A}_{21} \left( B(X_k) U_k + A_{uv}(X_k) Z_k + b_u(X_k) \right)$$
$$+ \epsilon^\beta \left( \tilde{B}(X_k) U_k + A_{vv}(X_k) Z_k + b_v(X_k) \right),$$

which can be written as a stochastic recursion in terms of $\Theta_k = (U_k, Z_k)$ as follows
$$\Theta_{k+1} = \Theta_k + \epsilon^\alpha \left( \tilde{A}(X_k) + \tilde{b}(X_k) \right), \tag{24}$$

where
$$\tilde{A}(X_k) = \begin{pmatrix} B(X_k) & A_{uv}(X_k) \\ \bar{A}_{vv}^{-1} \bar{A}_{vu} B(X_k) + \epsilon^{\beta-\alpha} \tilde{B}(X_k) & \bar{A}_{vv}^{-1} \bar{A}_{vu} A_{uv}(X_k) + \epsilon^{\beta-\alpha} A_{vv}(X_k) \end{pmatrix} \tag{25}$$
$$\tilde{b}(X_k) = \begin{pmatrix} b_u(X_k) \\ \bar{A}_{vv}^{-1} \bar{A}_{vu} b_u(X_k) + \epsilon^{\beta-\alpha} b_v(X_k) \end{pmatrix}. \tag{26}$$

We first establish a sequence of preliminary lemmas before we present the proof of Lemma 1.

**Lemma 2.** *For any $k \geq 0$, the following inequalities hold:*
$$\|\tilde{A}(X_k)\| \leq \delta,$$
$$\|\bar{\tilde{A}}\| \leq \delta,$$
$$\|\tilde{b}(X_k)\| \leq \delta b_{\max},$$
$$\bar{\tilde{b}} = 0,$$

*where $\delta = 2(1 + \|\bar{A}_{vv}^{-1} \bar{A}_{vu}\| + \epsilon^{\beta-\alpha})$, $\bar{\tilde{A}} = \lim_{k\to\infty} \tilde{A}(X_k)$, and $\bar{\tilde{b}} = \lim_{k\to\infty} \tilde{b}(X_k)$.*

*Proof.* We begin by proving the first inequality:
$$\begin{aligned}\|\tilde{A}(X_k)\| &\leq \|B(X_k)\| + \|A_{uv}(X_k)\| + \|\bar{A}_{vv}^{-1} \bar{A}_{vu} B(X_k)\| + \epsilon^{\beta-\alpha} \|\tilde{B}(X_k)\| \\ &\quad + \|\bar{A}_{vv}^{-1} \bar{A}_{vu} A_{uv}(X_k)\| + \epsilon^{\beta-\alpha} \|A_{vv}(X_k)\| \\ &\leq 1 + 1 + c + c + 2\epsilon^{\beta-\alpha} \\ &= 2(c + 1 + \epsilon^{\beta-\alpha}) \end{aligned} \tag{27}$$

where $c = \|\bar{A}_{vv}^{-1} \bar{A}_{vu}\|$ and the last inequality follows from the assumptions. Similarly, one can also show the remaining inequalities. $\square$

**Lemma 3.** *For $\Theta_\tau$ and $\Theta_0$, the following inequalities hold:*
$$\|\Theta_\tau - \Theta_0\| \leq 2\tilde{\epsilon}\tau \|\Theta_0\| + 2\tilde{\epsilon}\tau b_{\max}$$
$$\|\Theta_\tau - \Theta_0\| \leq 4\tilde{\epsilon}\tau \|\Theta_\tau\| + 4\tilde{\epsilon}\tau b_{\max}$$
$$\|\Theta_\tau - \Theta_0\|^2 \leq 32\tilde{\epsilon}^2\tau^2 \|\Theta_\tau\|^2 + 32\tilde{\epsilon}^2\tau^2 b_{\max}^2$$

*where $\tilde{\epsilon} = \epsilon^\alpha \delta$.*

*Proof.* Recall that $\delta = 2\left(1 + \|\bar{A}_{vv}^{-1}\bar{A}_{vu}\| + \frac{\epsilon^{\beta-\alpha}}{2}\right)$, therefore we have $\tilde{\epsilon} = \epsilon^\alpha \delta$. By applying Lemma 2, we obtain

$$\|\Theta_{k+1} - \Theta_k\| = \epsilon^\alpha \|\tilde{A}(X_k)\Theta_k + \tilde{b}(X_k)\| \leq \tilde{\epsilon}(\|\Theta_k\| + b_{\max}). \tag{28}$$

The result then follows from the steps in the proof of Lemma 3 in [Srikant and Ying, 2019]. $\qquad\square$

**Lemma 4.** *For any $k \geq 0$, the following inequality holds*

$$\left|(\Theta_{k+1} - \Theta_k)^\top P(\Theta_{k+1} - \Theta_k)\right| \leq 2\tilde{\epsilon}^2 \gamma_{\max}(\|\Theta_k\|^2 + b_{\max}^2).$$

*Proof.* The lemma follows directly from (28):

$$\begin{aligned}
\left|(\Theta_{k+1} - \Theta_k)^\top P(\Theta_{k+1} - \Theta_k)\right| &\leq \gamma_{\max}\|\Theta_{k+1} - \Theta_k\|^2 \\
&\leq \tilde{\epsilon}^2 \gamma_{\max}(\|\Theta_k\| + b_{\max})^2 \\
&\leq 2\tilde{\epsilon}^2 \gamma_{\max}(\|\Theta_k\|^2 + b_{\max}^2).
\end{aligned}$$

$\qquad\square$

**Lemma 5.** *For all $k \geq \tau$, the following inequality holds:*

$$\left|\mathbb{E}\left[\Theta_k^\top P\left(\bar{\bar{A}}\Theta_k - \frac{1}{\epsilon^\alpha}(\Theta_{k+1} - \Theta_k)\right)\middle|\Theta_{k-\tau}, X_{k-\tau}\right]\right|$$

$$\leq 10\tilde{\epsilon}\tau\gamma_{\max}(1 + 6\delta)(1 + b_{\max})\left(\mathbb{E}[\|\Theta_k\|^2|\Theta_{k-\tau}, X_{k-\tau}] + (1 + b_{\max})^2\right)$$

$$= \tilde{\eta}_1\tilde{\epsilon}\tau\mathbb{E}[\|\Theta_k\|^2|\Theta_{k-\tau}, X_{k-\tau}] + \tilde{\eta}_2\tilde{\epsilon}\tau.$$

*Proof.* For ease of notation, we prove the lemma for $k = \tau$, but the proof for any $k \geq \tau$ is identical. We consider

$$\begin{aligned}
&\mathbb{E}\left[\Theta_\tau^\top P\left(\bar{\bar{A}}\Theta_\tau - \frac{1}{\epsilon^\alpha}(\Theta_{\tau+1} - \Theta_\tau)\right)\middle|\Theta_0, X_0\right] \\
&= \mathbb{E}\left[\Theta_\tau^\top P\left(\bar{\bar{A}}\Theta_\tau - (\tilde{A}(X_\tau)\Theta_\tau + \tilde{b}(X_\tau))\right)\middle|\theta_0, X_0\right] \tag{29} \\
&= \mathbb{E}\left[\Theta_\tau^\top P\left(\bar{\bar{A}} - \tilde{A}(X_\tau)\right)\Theta_\tau\middle|\Theta_0, X_0\right] - \mathbb{E}\left[\Theta_\tau^\top P\tilde{b}(X_\tau)\middle|\Theta_0, X_0\right].
\end{aligned}$$

We first consider the first term on the RHS of the above equation:

$$\begin{aligned}
&\mathbb{E}\left[\Theta_\tau^\top P\left(\bar{\bar{A}} - \tilde{A}(X_\tau)\right)\Theta_\tau\middle|\Theta_0, X_0\right] \\
&= \mathbb{E}\left[\Theta_0^\top P\left(\bar{\bar{A}} - \tilde{A}(X_\tau)\right)\Theta_0\middle|\Theta_0, X_0\right] + \mathbb{E}\left[(\Theta_\tau - \Theta_0)^\top P\left(\bar{\bar{A}} - \tilde{A}(X_\tau)\right)(\Theta_\tau - \Theta_0)\middle|\Theta_0, X_0\right] \\
&\quad + \mathbb{E}\left[(\Theta_\tau - \Theta_0)^\top P\left(\bar{\bar{A}} - \tilde{A}(X_\tau)\right)\Theta_0\middle|\Theta_0, X_0\right] + \mathbb{E}\left[\Theta_0^\top P\left(\bar{\bar{A}} - \tilde{A}(X_\tau)\right)(\Theta_\tau - \Theta_0)\middle|\Theta_0, X_0\right].
\end{aligned}$$
$$\tag{30}$$

We will now analyze each term on the RHS above. Starting with the first term:

$$\begin{aligned}
\mathbb{E}\left[\Theta_0^\top P\left(\bar{\bar{A}} - \tilde{A}(X_\tau)\right)\Theta_0\middle|\Theta_0, X_0\right] &= \left|\Theta_0^\top P\left(\bar{\bar{A}} - \mathbb{E}[\tilde{A}(X_\tau)|X_0]\right)\Theta_0\right| \\
&\leq \left\|\Theta_0^\top P\right\|\left\|\left(\bar{\bar{A}} - \mathbb{E}[\tilde{A}(X_\tau)|X_0]\right)\Theta_0\right\|, \tag{31} \\
&\leq \tilde{\epsilon}\gamma_{\max}\|\Theta_0\|^2
\end{aligned}$$

where the final inequality follows from the assumptions on the mixing time $\tau$ and the fact that $\left\|\begin{pmatrix} \frac{1}{\bar{A}_{vv}^{-1}\bar{A}_{vu}} & \frac{1}{\bar{A}_{vv}^{-1}\bar{A}_{vu} + \epsilon^{\beta-\alpha}} \end{pmatrix}\right\| \leq \delta = 2(1 + \|\bar{A}_{vv}^{-1}\bar{A}_{vu}\| + \epsilon^{\beta-\alpha})$. Next, we bound the second term on the RHS of (30):

$$\begin{aligned}
&\left|\mathbb{E}\left[(\Theta_\tau - \Theta_0)^\top P\left(\bar{\bar{A}} - \tilde{A}(X_\tau)\right)(\Theta_\tau - \Theta_0)\middle|\Theta_0, X_0\right]\right| \\
&\leq \mathbb{E}\left[\|(\Theta_\tau - \Theta_0)^\top P\|\|\left(\bar{\bar{A}} - \tilde{A}(X_\tau)\right)(\Theta_\tau - \Theta_0)\|\middle|\Theta_0, X_0\right] \tag{32} \\
&\leq \gamma_{\max}\mathbb{E}\left[(\|\bar{\bar{A}}\| + \|\tilde{A}(X_\tau)\|)\|\Theta_\tau - \Theta_0\|^2\middle|\Theta_0, X_0\right] \\
&\leq 2\delta\gamma_{\max}\mathbb{E}\left[\|\Theta_\tau - \Theta_0\|^2|\Theta_0, X_0\right]
\end{aligned}$$

where the last inequality follows from Lemma 2. Finally, we bound the third and fourth terms on the RHS of (30):

$$\left| \mathbb{E}\big[(\Theta_\tau - \Theta_0)^\top P(\bar{\bar{A}} - \tilde{A}(X_\tau))\Theta_0 | \Theta_0, X_0\big] \right| + \left| \mathbb{E}\big[\Theta_0^\top P(\bar{\bar{A}} - \tilde{A}(X_\tau))(\Theta_\tau - \Theta_0) | \Theta_0, X_0\big] \right|$$

$$\leq 4\delta\gamma_{\max}\|\Theta_0\|\mathbb{E}[\|\Theta_\tau - \Theta_0\| | \Theta_0, X_0]$$
$$\leq 8\tilde{\epsilon}\delta\tau\gamma_{\max}\|\Theta_0\|(\|\Theta_0\| + b_{\max})$$
$$\leq 8\tilde{\epsilon}\delta\tau\gamma_{\max}\|\Theta_0\|^2 + 8\epsilon'\delta\tau\gamma_{\max}\|\Theta_0\|b_{\max}$$

(33)

where the first inequality follows from Lemma 2 and the second inequality follows from Lemma 3.

Next we consider the second term on the RHS of (29):

$$\left| -\mathbb{E}\big[\Theta_\tau^\top P\tilde{b}(X_\tau) | \Theta_0, X_0\big] \right|$$

$$= \left| -\mathbb{E}\big[\Theta_0^\top P\tilde{b}(X_\tau) | \Theta_0, X_0\big] - \mathbb{E}\big[(\Theta_\tau - \Theta_0)^\top P\tilde{b}(X_\tau) | \Theta_0, X_0\big] \right|$$

(34)

$$\leq \tilde{\epsilon}\gamma_{\max}\|\Theta_0\| + \gamma_{\max}b_{\max}\mathbb{E}[\|\Theta_\tau - \Theta_0\| | \Theta_0, X_0]$$
$$\leq \tilde{\epsilon}\gamma_{\max}\|\Theta_0\| + 2\tilde{\epsilon}\tau\gamma_{\max}b_{\max}(\|\Theta_0\| + b_{\max})$$

where the final inequality follows from Lemma 3.

Now, combining (31) - (34), we get

$$\left| \mathbb{E}\big[\Theta_k^\top P\left(\bar{\bar{A}}\Theta_k - \frac{1}{\epsilon^\alpha}(\Theta_{k+1} - \Theta_k)\right) | \Theta_{k-\tau}, X_{k-\tau}\big] \right|$$

$$\leq \big(\tilde{\epsilon}\gamma_{\max} + 8\tilde{\epsilon}\delta\tau\gamma_{\max}\big)\|\Theta_0\|^2 + 2\tilde{\epsilon}\tau\gamma_{\max}b_{\max}^2$$
$$\quad + \big(8\tilde{\epsilon}\delta\tau\gamma_{\max}b_{\max} + \tilde{\epsilon}\gamma_{\max} + 2\tilde{\epsilon}\tau\gamma_{\max}b_{\max}\big)\|\Theta_0\|$$
$$\quad + 2\delta\gamma_{\max}\mathbb{E}[\|\Theta_\tau - \Theta_0\|^2 | \Theta_0, X_0]$$
$$\leq \big(2\tilde{\epsilon}\gamma_{\max} + 8\tilde{\epsilon}\delta\tau\gamma_{\max} + \tilde{\epsilon}\tau\gamma_{\max}b_{\max} + 4\tilde{\epsilon}\delta\tau\gamma_{\max}b_{\max}\big)\|\Theta_0\|^2$$
$$\quad + 2\tilde{\epsilon}\tau\gamma_{\max}b_{\max} + 4\tilde{\epsilon}\delta\tau\gamma_{\max}b_{\max} + \tilde{\epsilon}\gamma_{\max} + 2\tilde{\epsilon}\tau\gamma_{\max}b_{\max}^2$$
$$\quad + 2\delta\gamma_{\max}\mathbb{E}[\|\Theta_\tau - \Theta_0\|^2 | \Theta_0, X_0]$$
$$\leq \big(2\tilde{\epsilon}\tau\gamma_{\max}(1 + 4\delta)(1 + b_{\max})\big)\|\Theta_0\|^2 + \tilde{\epsilon}\tau\gamma_{\max}\big((2b_{\max} + 1)^2 + 4\delta b_{\max}\big)$$
$$\quad + 2\delta\gamma_{\max}\mathbb{E}[\|\Theta_\tau - \Theta_0\|^2 | \Theta_0, X_0]$$

(35)

$$\leq \big(2\tilde{\epsilon}\tau\gamma_{\max}(1 + 4\delta)(1 + b_{\max})\big)\mathbb{E}[\|\Theta_\tau\|^2 | \Theta_0, X_0]$$
$$\quad + \tilde{\epsilon}\tau\gamma_{\max}\big((2b_{\max} + 1)^2 + 4\delta b_{\max}\big)$$
$$\quad + \big(\gamma_{\max}(1 + 6\delta)(1 + b_{\max})\big)\mathbb{E}[\|\Theta_\tau - \Theta_0\|^2 | \Theta_0, X_0]$$
$$\leq \big(2\tilde{\epsilon}\tau\gamma_{\max}(1 + 4\delta)(1 + b_{\max})\big)\mathbb{E}[\|\Theta_\tau\|^2 | \Theta_0, X_0]$$
$$\quad + \tilde{\epsilon}\tau\gamma_{\max}\big((2b_{\max} + 1)^2 + 4\delta b_{\max}\big)$$
$$\quad + \big(\gamma_{\max}(1 + 6\delta)(1 + b_{\max})\big)\big(32\tilde{\epsilon}^2\tau^2\mathbb{E}[\|\Theta_\tau\|^2 | \Theta_0, X_0] + 32\tilde{\epsilon}^2\tau^2 b_{\max}^2\big)$$
$$\leq 10\tilde{\epsilon}\tau\gamma_{\max}(1 + 6\delta)(1 + b_{\max})\mathbb{E}[\|\Theta_\tau\|^2 | \Theta_0, X_0]$$
$$\quad + 10\tilde{\epsilon}\tau\gamma_{\max}(1 + 6\delta)(1 + b_{\max})^3$$

where the second inequality follows from the fact that $2\|\theta_0\| \leq 1 + \|\theta_0\|^2$ and $\tau \geq 1$, the fourth inequality follows from the triangle inequality and the penultimate inequality follows from Lemma 3. $\qquad\square$

Next we lower bound the minimum eigenvalue of the matrix-valued function $\Psi(\cdot)$.

**Lemma 6.** *Let* $\Psi(\mu) = \begin{pmatrix} \frac{\xi_2}{\xi_1+\xi_2} & -\frac{\xi_1\xi_2}{\xi_1+\xi_2} \\ -\frac{\xi_1\xi_2}{\xi_1+\xi_2} & \frac{1}{\mu}\frac{\xi_1}{\xi_1+\xi_2} - \frac{\mu\nu\xi_1}{\xi_1+\xi_2} \end{pmatrix}$ *with* $\xi_1, \xi_2, \nu > 0$ *and* $\mu \geq 0$. *Then, the following holds*

$$\lambda_{\min}(\Psi(\mu)) \geq \kappa_1 - \kappa_2\mu$$

*where $\kappa_1 = \frac{\xi_2}{\xi_1 + \xi_2}$ and $\kappa_2$ is a constant that depends only on $\xi_1, \xi_2$ and $\nu$.*

*Proof.* The minimum eigenvalue of a $2 \times 2$ matrix $\begin{pmatrix} a & b \\ c & d \end{pmatrix}$ is

$$\frac{1}{2}(a + d - \sqrt{(a-d)^2 + 4bc}),$$

so we have

$$\lambda_{\min}(\Psi(\mu)) = \frac{1}{2} \left( \frac{\xi_2}{\xi_1 + \xi_2} + \frac{\xi_1}{\xi_1 + \xi_2}(\frac{1}{\mu} - \nu) \right. \tag{36}$$
$$\left. - \frac{\xi_1}{\xi_1 + \xi_2} \sqrt{(\frac{\xi_2}{\xi_1} - (\frac{1}{\mu} - \nu))^2 + (2\xi_2)^2} \right)$$

In order to obtain a lower bound on $\lambda_{\min}(\Psi(\mu))$, we first establish an upper bound on the third term on the RHS in the above equation. Defining $f(\mu) = \mu \sqrt{(\frac{\xi_2}{\xi_1} - (\frac{1}{\mu} - \nu))^2 + (2\xi_2)^2}$, we have

$$f'(0) = -(\nu + \frac{\xi_2}{\xi_1})$$
$$f''(\mu) = \frac{(\nu + \frac{\xi_2}{\xi_1})^2 + 4\xi_2^2}{f(\mu)^2} - \frac{f'(\mu)^2}{f(\mu)} \tag{37}$$
$$\leq \max_{\mu \geq 0} \frac{(\nu + \frac{\xi_2}{\xi_1})^2 + 4\xi_2^2}{f(\mu)^2} - \frac{f'(\mu)^2}{f(\mu)} = 2\kappa_2 < \infty$$

which implies that

$$f(\mu) \leq f(0) + f'(0)\mu + \kappa_2 \mu^2$$
$$= 1 - (\nu + \frac{\xi_2}{\xi_1})\mu + \kappa_2 \mu^2. \tag{38}$$

Substituting the above equation into (36) yields

$$\lambda_{\min}(\Psi(\mu)) \geq \frac{1}{2} \left( \frac{\xi_2}{\xi_1 + \xi_2} + \frac{\xi_1}{\xi_1 + \xi_2}(\frac{1}{\mu} - \nu) - \frac{1}{\mu} \frac{\xi_1}{\xi_1 + \xi_2} \left( 1 - (\nu + \frac{\xi_2}{\xi_1})\mu + \kappa_2 \mu^2 \right) \right)$$
$$\geq \frac{1}{2} \left( \frac{2\xi_1 2}{\xi_1 + \xi_2} - 2\kappa_2 \mu \right) \tag{39}$$
$$= \kappa_1 - \kappa_2 \mu$$

$\square$

We are now ready to prove Lemma 1. For any $k \geq \tau$, we have:

$$\mathbb{E}\left[ W(\Theta_{k+1}) - W(\Theta_k) | \Theta_{k-\tau}, X_{k-\tau} \right]$$
$$= \mathbb{E}\left[ 2\Theta_k^\top P(\Theta_{k+1} - \Theta_k) + (\Theta_{k+1} - \Theta_k)^\top P(\Theta_{k+1} - \Theta_k) | \Theta_{k-\tau}, X_{k-\tau} \right]$$
$$= \mathbb{E}[2\Theta_k^\top P(\Theta_{k+1} - \Theta_k - \epsilon^\alpha \bar{A}\Theta_k) + (\Theta_{k+1} - \Theta_k)^\top P(\Theta_{k+1} - \Theta_k) | \Theta_{k-\tau}, X_{k-\tau}]$$
$$+ 2\epsilon^\alpha \mathbb{E}[\Theta_k^\top P\bar{A}\Theta_k | \Theta_{k-\tau}, X_{k-\tau}].$$

Using the facts that $P_u$ and $P_v$ are the solutions to their respective Lyapunov equations, we have

$$\mathbb{E}\left[ \Theta_k^\top P\bar{A}\Theta_k | \Theta_{k-\tau}, X_{k-\tau} \right] \leq -\lambda_{\min}\mathbb{E}\left[ \|\Theta_k\|^2 | \Theta_{k-\tau}, X_{k-\tau} \right] \tag{40}$$

where $\lambda_{\min}$ is the smallest eigenvalue of

$$\Psi = \frac{1}{\xi_1 + \xi_2} \begin{pmatrix} \xi_2 & -\xi_1 \xi_2 \\ -\xi_1 \xi_2 & \xi_1 \left( \epsilon^{-\alpha+\beta} - 2\|P_v \bar{A}_{vv}^{-1} \bar{A}_{vu} \bar{A}_{uv}\| \right) \end{pmatrix}.$$

Combining the above equation, Lemma 4 and Lemma 5 with (40), we obtain

$$\mathbb{E}[W(\Theta_{k+1}) - W(\Theta_k)|\Theta_{k-\tau}, X_{k-\tau}]$$
$$\leq -2\epsilon^\alpha \lambda_{\min}\mathbb{E}[\|\Theta_k\|^2|\Theta_{k-\tau}, X_{k-\tau}]$$
$$+ \epsilon^\alpha\left(\tilde{\eta}_1\tilde{\epsilon}\tau\mathbb{E}[\|\Theta_k\|^2|\Theta_{k-\tau}, X_{k-\tau}] + \tilde{\eta}_2\tilde{\epsilon}\tau\right) + 2\tilde{\epsilon}^2\gamma_{\max}\left(\mathbb{E}[\|\Theta_k\|^2|\Theta_{k-\tau}, X_{k-\tau}] + b_{\max}^2\right)$$
$$\leq \mathbb{E}[\|\Theta_k\|^2|\Theta_{k-\tau}, X_{k-\tau}]\left(-2\epsilon^\alpha\lambda_{\min} + \tilde{\eta}_1\epsilon^\alpha\tilde{\epsilon}\tau + 2\tilde{\epsilon}^2\gamma_{\max}\right)$$
$$+ \epsilon^\alpha\tilde{\epsilon}\tau\left(\tilde{\eta}_2 + 4\left(1 + \|\bar{A}_{vv}^{-1}\bar{A}_{vu}\| + \epsilon^{\beta-\alpha}\right)\right).$$

Applying the bound on $\lambda_{\min}$ in Lemma 6, we further get

$$\mathbb{E}[W(\Theta_{k+1}) - W(\Theta_k)|\Theta_{k-\tau}, X_{k-\tau}]$$
$$\leq \mathbb{E}[\|\Theta_k\|^2|\Theta_{k-\tau}, X_{k-\tau}]\left(-\epsilon^\alpha(\kappa_1 - \kappa_2\epsilon^{\alpha-\beta}) + \tilde{\eta}_1\epsilon^\alpha\tilde{\epsilon}\tau + 2\tilde{\epsilon}^2\gamma_{\max}\right)$$
$$+ \epsilon^\alpha\tilde{\epsilon}\tau\left(\tilde{\eta}_2 + 4\left(1 + \|\bar{A}_{vv}^{-1}\bar{A}_{vu}\| + \epsilon^{\beta-\alpha}2\right)\right)$$
$$\leq \mathbb{E}\left[\|\Theta_k\|^2|\Theta_{k-\tau}, X_{k-\tau}\right]\left(-\epsilon^\alpha\left(\frac{\kappa_1}{2} - \kappa_2\epsilon^{\alpha-\beta}\right)\right) + \epsilon^\alpha\tilde{\epsilon}\tau\left(\tilde{\eta}_2 + 4\left(1 + \|\bar{A}_{vv}^{-1}\bar{A}_{vu}\| + \epsilon^{\beta-\alpha}\right)\right)$$
$$\leq \mathbb{E}\left[\|\Theta_k\|^2|\Theta_{k-\tau}, X_{k-\tau}\right]\left(-\epsilon^\alpha\left(\frac{\kappa_1}{2} - \kappa_2\epsilon^{\alpha-\beta}\right)\right)$$
$$+ \epsilon^{2\beta}\tau\left((3 + 2\|\bar{A}_{vv}^{-1}\bar{A}_{vu}\|)(\tilde{\eta}_2 + 4(1 + \|\bar{A}_{vv}^{-1}\bar{A}_{vu}\|)) + 6 + 4\|\bar{A}_{vv}^{-1}\bar{A}_{vu}\|\right)$$
$$= \mathbb{E}\left[\|\Theta_k\|^2|\Theta_{k-\tau}, X_{k-\tau}\right]\left(-\epsilon^\alpha\left(\frac{\kappa_1}{2} - \kappa_2\epsilon^{\alpha-\beta}\right)\right) + \epsilon^{2\beta}\tau\eta_2$$
$$\leq -\frac{\epsilon^\alpha}{\gamma_{\max}}\left(\frac{\kappa_1}{2} - \kappa_2\epsilon^{\alpha-\beta}\right)\mathbb{E}[W(\Theta_k)] + \epsilon^{2\beta}\tau\eta_2,$$

$$(41)$$

where the second inequality follows from the assumption on $\epsilon$, $\alpha$ and $\beta$, and the third inequality follows from the fact that $\epsilon < 1$ and $\alpha > \beta$.

## B  The Lyapunov function (7)

The rationale behind the Laypunov function is well known to control theorists, but we present it here for the interested reader.

- Setting $\epsilon = 0$ in (6) is equivalent to studying the system of ODEs in a slow time-scale where the fast time-scale dynamics are assumed to converge instantaneously. In this case, for a fixed $u$, $v$ can be written as $v_u = -\bar{A}_{vv}^{-1}\bar{A}_{vu}u$ and substituting this expression in (5), the ODE is purely in terms of $u$. The first term $u^T P_u u$ in (7) is the standard Lyapunov function used in control theory to study the stability of the resulting ODE for $u$.

- The second term $\left(v + \bar{A}_{vv}^{-1}\bar{A}_{vu}u\right)^\top P_v \left(v + \bar{A}_{vv}^{-1}\bar{A}_{vu}u\right)$ studies the convergence of $v$ to $v_u$ for a fixed $u$ and thus, corresponds to the stability of the fast subsystem.

## C  Experimental Setup Details

Following is a detailed description of reinforcement learning problems/domains we implemented[1]:

1. **Mountain Car:** In the basic mountain car problem, an underpowered car is positioned in a valley between two mountains on a one-dimensional track. The aim of the problem is to drive the car to the top of the mountain on the right-hand side, but the engine power available is insufficient to simply accelerate and power through to the top. Therefore, a player has to build up momentum by going back and forth between the two mountains until the car has sufficient momentum to reach its goal. The state space, action space, cost structure and initialization details for the mountain car problem are as follows:

    - *State Space:* (Car Position, Car Velocity) $\in [-1.2, 0.6] \times [-0.07, 0.07]$.

- *Action Space:* 0, 1 and 2 (denoting left, no and right acceleration respectively).
- *Cost Structure:* +1 cost incurred for every time step the car has not achieved its goal. 0 cost incurred upon reaching the goal.
- *Initialization/Starting State:* The car's position is initialized to a random value in $[-0.6, 0.4]$. Its velocity is initialized to 0.

2. **Inverted Pendulum:** In the classic inverted pendulum swing-up problem, a frictionless pendulum is hinged/pivoted at one end and the aim of the problem is to keep the pendulum in an upright position (with respect to the pivot) for as long as possible by applying a torque at the pivot point (sometimes referred to as the joint effort). The state space, action space, cost structure and initialization details for the inverted pendulum problem are as follows:

- *State Space:* $(\cos(\theta), \sin(\theta), \dot{\theta}) \in [-1.0, 1.0] \times [-1.0, 1.0] \times [-8.0, 8.0]$. Here, $\theta \in [-\pi, \pi]$ denotes the angular position of the pendulum with respect to the pivot.
- *Action Space:* Torque $\in [-2.0, 2.0]$.
- *Cost Structure:* The equation associated with the cost function is the following:

$$-(\theta^2 + 0.1\dot{\theta} + 0.001 \times \text{torque}^2).$$

- *Initialization/Starting State:* The pendulum's angular position is initialized to a random value in $[-\pi, \pi]$. Its angular velocity is initialized to a random value $\in [-1, 1]$.

## D  Slope Calculations

### D.1  Bounding $\mathbb{E}[\|\psi_N\|]$

We have the following $N$ points: $\{X_i = i, Y_i = \|\Theta_{k+i} - \Theta_0\|\}_{i=1}^N$. Using the formula for the slope of the best-fit line passing through these points, we get:

$$\psi_N = \frac{\sum_{i=1}^N (X_i - \bar{X})(Y_i - \bar{Y})}{\sum_{i=1}^N (X_i - \bar{X})^2} \tag{42}$$

where $\bar{X} = \frac{\sum_{i=1}^N X_i}{N} = \frac{1}{N}\sum_{i=1}^N X_i = \frac{N+1}{2}$ and $\bar{Y} = \frac{\sum_{i=1}^N Y_i}{N}$. Also, note that $\sum_{i=1}^N (X_i - \bar{X})^2 = \sum_{i=1}^N (i - \frac{N+1}{2})^2 = \frac{N(N-1)(N+1)}{12}$. Therefore, we have

$$\mathbb{E}[\psi_N] = \frac{12 \sum_{i=1}^N (i - \frac{N+1}{2})\mathbb{E}[(Y_i - \bar{Y})]}{N(N-1)(N+1)} \tag{43}$$

From (22) we know that $d - \sqrt{\frac{2K_2\mu^{2-\lambda}}{\gamma_{\max}c}} \leq \mathbb{E}[Y_i] \leq d + \sqrt{\frac{2K_2\mu^{2-\lambda}}{\gamma_{\max}c}}$. This also implies that $d - \sqrt{\frac{2K_2\mu^{2-\lambda}}{\gamma_{\max}c}} \leq \mathbb{E}[\bar{Y}] \leq d + \sqrt{\frac{2K_2\mu^{2-\lambda}}{\gamma_{\max}c}}$. Using these two facts in (43)

$$|\mathbb{E}[\psi_N]| \leq \frac{24 \left( \sum_{i=1}^{\lfloor \frac{N+1}{2} \rfloor} (\frac{N+1}{2} - i) + \sum_{i=\lfloor \frac{N+1}{2} \rfloor + 1}^N (i - \frac{N+1}{2}) \right) \sqrt{\frac{2K_2\mu^{2-\lambda}}{\gamma_{\max}c}}}{N(N-1)(N+1)}$$

$$\leq \frac{24 \left( \sum_{i=1}^{\lfloor \frac{N+1}{2} \rfloor} (\frac{N+1}{2} - i) + \sum_{i=1}^{N - \lfloor \frac{N+1}{2} \rfloor} i \right) \sqrt{\frac{2K_2\mu^{2-\lambda}}{\gamma_{\max}c}}}{N(N-1)(N+1)}$$

$$\leq \frac{24 \frac{(N+1)^2}{4} \sqrt{\frac{2K_2\mu^{2-\lambda}}{\gamma_{\max}c}}}{N(N-1)(N+1)} = O\left( \frac{\mu^{1-\frac{\lambda}{2}}}{N} \right)$$

where the second inequality follows from centering the second summation term in the numerator and the last inequality follows from the fact that $\sum_{i=1}^{\lfloor \frac{N+1}{2} \rfloor} -i + \sum_{i=1}^{N-\lfloor \frac{N+1}{2} \rfloor} i \leq 0$.

## D.2 Bounding Var$(\psi_N)$

Using (42):

$$
\begin{aligned}
\mathbb{E}[\psi_N^2] &= \frac{\mathbb{E}[\left(\sum_{i=1}^N (X_i - \bar{X})(Y_i - \bar{Y})\right)^2]}{\left(\sum_{i=1}^N (X_i - \bar{X})^2\right)^2} \\
&\leq \frac{\sum_{i=1}^N (X_i - \bar{X})^2 \mathbb{E}[\sum_{i=1}^N (Y_i - \bar{Y})^2]}{\left(\sum_{i=1}^N (X_i - \bar{X})^2\right)^2} \\
&= \frac{\mathbb{E}[\sum_{i=1}^N (Y_i - \bar{Y})^2]}{\sum_{i=1}^N (X_i - \bar{X})^2} \\
&\leq \frac{24\mathbb{E}[\sum_{i=1}^N (Y_i^2 + \bar{Y}^2)]}{N(N-1)(N+1)} \\
&\leq \frac{24\mathbb{E}[\sum_{i=1}^N (Y_i^2 + \frac{\sum_{i=1}^N Y_i^2}{N})]}{N(N-1)(N+1)} \\
&\leq \frac{48\left(\frac{4K_2\mu^{2-\lambda}}{\gamma_{\max} c} + 2\|\Theta_0 - \Theta^*\|^2+\right)}{(N-1)(N+1)} = O(\frac{1}{N^2})
\end{aligned}
\tag{44}
$$

where the first inequality follows from the Cauchy-Schwarz inequality, the second inequality follows from the fact that $(a+b)^2 \leq 2a^2 + 2b^2$ and $\sum_{i=1}^N (X_i - \bar{X})^2 = \sum_{i=1}^N (i - \frac{N+1}{2})^2 = \frac{N(N-1)(N+1)}{12}$, the third inequality follows from Cauchy-Schwarz inequality and the final inequality follows from (20) - (21) and the fact that $(a+b)^2 \leq 2a^2 + 2b^2$.

## Footnotes

[1]We used the OpenAI Gym implementation of these environments, available at https://gym.openai.com/.