[Reviews · NeurIPS 2019]

Reviewer 1



Originality: Though it seems that the proof for bounds obtained in the paper follows along the lines of [Srikant & Ying, 2019], I don't think the extension is trivial. The key difference being the construction of a Lyapunov function that is motivated by singular perturbation theory. The second contribution on an algorithm that adaptively schedules the learning rate is novel, and not something I have seen before. Quality and Clarity: Except for minor typos, the paper was well-written, and easy to read. Specifically, the intuition for Lemma 1 by studying the associated ODE's was very useful. Significance: As stated above, the paper provides good foundations as well as directions for future research. ** After reading the rebuttal ** The authors addressed my concerns well, and I would like to still recommend acceptance of the paper.

Reviewer 2



Nice paper, and very clear presentation of the main results. Here are few suggestions for expanding your presentation: 1. The literature review needs to be broadened. In particular, you should discuss the work of Liu et al., JAIR 2019 (Proximal Gradient TD Learning), which analyzes two time-scale algorithms that include a proximal gradient step. The results in that paper show improved finite sample bounds over classic gradient TD methods, like GTD2. How do your results compare with those in that paper, and in particular, can your analysis be extended to GTD2-MP (the mirror-prox variant of GTD2, which has an improved finite sample convergence rate compared to GTD2. 2. Two time scale algorithms are somewhat more complex than the standard TD method, and Sutton et al. and others have developed a variant of TD called emphatic TD (JMLR 2016: "An Emphatic Approach to the Problem of Off-policy Temporal-Difference Learning") that is stable under off-policy training. How does your analysis relate to emphatic TD methods? 3. Your analysis is largely set in the context of linear function approximation, but of course, all the recent excitement in RL is over deep nonlinear function approximation networks. Does your learning rate adaptation scheme apply to nonlinear deep neural networks and have you done any experiments on such networks? 4. The presentation can be improved. Some of the main theoretical results (e.g., Theorem 1) would benefit from some simpler exposition. Rather than just state the exact theorem, it would help to add a sentence or two distilling the main implication into easier to parse language for those who want to get a gist of the main result.

Reviewer 3



**Things to like** I like this paper. 1. Quality. First of all, it is well written and pleasant to read. I couldn’t find any grammatical mistakes. 2. Significance. The authors use their analysis to produce something useful: a heuristic for learning rate selection. Then they provide some experimental validation that this heuristic is useful. I appreciate seeing this in a theory paper. 3. Quality. The technical aspects of the experiments are solid. **Concerns** Take these concerns with a grain of salt. I set my confidence score to 2, as I have a Sutton & Barto-level knowledge of reinforcement learning theory. 1. Clarity. I am worried about how the amount of mathematical notation affects clarity. The authors are not obfuscating anything; I believe this topic simply involves a lot of math. However, I doubt if the majority of NeurIPS attendees will be able to understand this paper. Since there are previous RL theory papers accepted to NeurIPS with this density of math [2], I don’t believe this is a fatal flaw and I did not take this into account in my overall score. 2. Significance. As a practitioner of RL, I’d like the authors to provide a more convincing argument about why I should care about this analysis. I’m not saying I don’t care (I’m quite interested), but I would love to hear the authors make a case for why and how this work affects RL “experimentalists.” [1] Dalal, Gal, et al. "Finite sample analysis of two-timescale stochastic approximation with applications to reinforcement learning." arXiv preprint arXiv:1703.05376 (2017). [2] Hasselt, Hado V. "Double Q-learning." Advances in Neural Information Processing Systems. 2010.

[Author Response · NeurIPS 2019]

We thank all the reviewers for their encouraging comments.

**Response to Reviewer 1:**

1. We can give bounds for the case of $k \leq \tau$ but such bounds are not very insightful since they scale linearly with $\tau$ (see
Lemma 3 in our supplementary material and Theorem 3 in Bhandari et al., 2018 - https://bit.ly/2YpL5Bm). Moreover,
in prior work (e.g., Bhandari et al, 2018), when there are bounds for $k \geq 0$, the underlying assumption is either that the
noise is i.i.d. or, if it is Markovian, it is in steady state starting at $k = 0$. In both these cases, $\tau$ is effectively zero.

2. The adaptive learning rate rule does reduce the step-size, but the decay is not periodic. The rule first tests whether the
algorithm is in steady-state, and if so, then the step-size is reduced. The number of iterations to reach steady-state will
be different for different step-sizes and hence the decay will not be periodic. We do expect the rule to converge to the
optimal parameter vector, although a proof of this result is a direction for further research.

3. The Konda-Tsitsiklis paper suggested by the reviewer seems to assume i.i.d noise, but it is an important reference
and we will cite it. We would like to mention that we had implemented the simulations using $\alpha = 1$ and $\beta = \frac{2}{3}$, but we
reported the results for $\alpha = 0.99$ and $\beta = 0.66$ since Dalal et al., 2017 shows that $\alpha \to 1$ and $\beta \to \frac{2}{3}$ are optimal for
the polynomial decay step-size rule. Moreover, $\alpha = 1, \beta = \frac{2}{3}$ had almost the same performance as $\alpha = 0.99, \beta = 0.66$.
Nevertheless, for the sake of completeness, in the final version of the paper, we will include experiments in which we
use $\alpha = 1$ and choose the best $\beta$ by conducting an extensive grid search.

4. The complexity of scheduling the step-sizes in Algorithm 1 depends on the implementation of linear regression to
compute the slope. In practice, instead of using consecutive points to compute the slope, one can use every $m^{th}$ point to
reduce the amount of computation and storage. Further, Recursive Least Squares can be potentially used to further
reduce the computational complexity.

**Response to Reviewer 2:**

1 and 2. We thank the reviewer for bringing the two papers (Liu et al., JAIR 2019 and Sutton et al., JMLR 2016) to
our attention. Liu et al. shows how GTD-class algorithms can be formally derived using a primal-dual saddle point
objective function. They use insights from this derivation to provide finite-time guarantees for GTD-class algorithms
with linear function approximation and design a more efficient algorithm called GTD2-MP. However, the authors restrict
their analysis and results to single time-scale GTD algorithms (see Algorithms 1 and 2). In fact, in Section 5.4, they
discuss how their analysis cannot be extended to TDC because TDC is a two time-scale algorithm (no single time-scale
version). In addition to analyzing only single time-scale algorithms, the paper also assumes an additional projection
step and i.i.d. sampling of data points from the steady-state distribution.

Sutton et al. presents a (single time-scale) variant of linear TD learning, which they call emphatic TD and show that
it is stable under off-policy training unlike standard linear TD. As mentioned in our submission (Section 4, lines
209-226), although we present our analysis for two time-scale algorithms, it is more general and when specialized to the
single-time scale, it will recover the finite-time guarantees in Srikant-Ying (COLT, 2019). These finite-time guarantees
will apply to both emphatic TD as well as GTD2-MP, since they are both single time-scale, and further improve upon
prior work since we do not assume an additional projection step or i.i.d. noise.

3. Maei et al., 2009 (link - https://bit.ly/32P3hrw) proposes generalizations of GTD2 and TDC to nonlinear arbitrary
smooth function approximation. They also provide an asymptotic convergence analysis to the set of local optima. We
are currently trying to extend our analysis to these generalizations in order to gain insight into using these algorithms
with deep neural networks. Once the analysis is complete, we will use it to further refine our adaptive learning rate rule
and apply it in experiments using nonlinear deep neural networks.

4. We will add concise and accessible explanations of the main theoretical results in the final version of the paper.

**Response to Reviewer 3:**

1. If the paper is accepted, we will work further on improving the clarity of the work. Specifically, we will try to make
the theoretical results more accessible to readers who do not have strong mathematical familiarity with RL. In particular,
we will present more intuition regarding the stability theory for singularly perturbed ODEs and how it relates to the
model and proofs in our paper.

2. The main theoretical result in the paper provides an insight into the role of different parameters ($\epsilon, \alpha, \beta$ and $\tau$) in
determining the rate of convergence of a two time-scale algorithm. For a practitioner, this result can be of immense
utility in designing sample-efficient two time-scale RL algorithms, since the rate of convergence can be better optimized
with the knowledge of the impact of different parameters. The adaptive learning rate rule that we have designed is
one such illustration of using the result to make it directly useful to an experimentalist. Further practically useful
improvements to the convergence rate may be possible using the theory in this paper, and we are currently investigating
such ideas.

[Meta-Review · NeurIPS 2019]

The reviewers unanimously support acceptance. We encourage the authors to strongly consider the suggestions provided by the reviewers for improving a camera ready version.